# Interpretable brain age prediction using linear latent variable models of functional connectivity

Ricardo Pio Monti[1,7]*, Alex Gibberd[2], Sandipan Roy[3], Matthew Nunes[3], Romy Lorenz[4,5], Robert Leech[6], Takeshi Ogawa[8], Motoaki Kawanabe[7,8], Aapo Hyvärinen[9,10]

**1** Gatsby Computational Neuroscience Unit, University College London, London, United Kingdom, **2** Department of Mathematics & Statistics, Lancaster University, Bailrigg, United Kingdom, **3** Department of Mathematical Sciences, University of Bath, Bath, United Kingdom, **4** MRC Cognition and Brain Sciences Unit, University of Cambridge, Cambridge, United Kingdom, **5** Department of Psychology, Stanford University, Stanford, CA, United States of America, **6** Centre for Neuroimaging Science, Kings College London, London, United Kingdom, **7** RIKEN Center for Advanced Intelligence Project (AIP), Kyoto, Japan, **8** Brain Information Communication Research Laboratory Group, Advanced Telecommunications Research Institute International (ATR), Kyoto, Japan, **9** Université Paris-Saclay, Inria, 91190 Palaiseau, France, **10** Department of Computer Science and HIIT, University of Helsinki, Helsinki, Finland

* ricardo.monti08@gmail.com

**Data Availability Statement:** With respect to the CamCAN data, the resting state fMRI data was employed. This can be accessed at: https://camcan-archive.mrc-cbu.cam.ac.uk/dataaccess/.

## Abstract

Neuroimaging-driven prediction of brain age, defined as the predicted biological age of a subject using only brain imaging data, is an exciting avenue of research. In this work we seek to build models of brain age based on functional connectivity while prioritizing model interpretability and understanding. This way, the models serve to both provide accurate estimates of brain age as well as allow us to investigate changes in functional connectivity which occur during the ageing process. The methods proposed in this work consist of a two-step procedure: first, linear latent variable models, such as PCA and its extensions, are employed to learn reproducible functional connectivity networks present across a cohort of subjects. The activity within each network is subsequently employed as a feature in a linear regression model to predict brain age. The proposed framework is employed on the data from the CamCAN repository and the inferred brain age models are further demonstrated to generalize using data from two open-access repositories: the Human Connectome Project and the ATR Wide-Age-Range.

## 1 Introduction

The human brain changes during the lifespan of an adult, resulting in robust and reproducible changes in structure and function [1, 2]. Moreover, there is reason to hypothesize that deviations from the typical brain ageing trajectory may reflect latent neuropathological influences [3], serving to motivate further research into developing reliable biomarkers derived from brain imaging data. Such biomarkers could be fundamental in order to better understand and

With respect to the HCP data, we studied resting state fMRI data from HCP Young Adult dataset: https://www.humanconnectome.org/study/hcp-young-adult/document/1200-subjects-data-release. With respect to the ATR Wide-Age-Range data, the resting state fMRI data was studied: https://bicr-resource.atr.jp/impact/.

**Funding:** R.P.M. was supported by the Gatsby Charitable Foundation. A.H. was supported by a Fellowship from CIFAR, and from the DATAIA convergence institute as part of the "Programme d'Investissement d'Avenir", (ANR-17-CONV-0003) operated by Inria. M.K. was partially supported by MEXT Grant-in-Aid for Scientific Research (KAKENHI 18KK0284, 19H04924).

**Competing interests:** The authors have declared that no competing interests exist.

combat age-associated neurodegenerative diseases. To date, early studies have shown success in the context of traumatic brain injury [4] and schizophrenia [5].

Due to the significant potential benefits associated with brain-imaging driven biomarkers for age, there have been many statistical models proposed for healthy brain ageing. These models vary in complexity as well as in the class of neuroimaging data employed. One of the earliest demonstrations was that of [6], who employed voxel-based morphometry to demonstrate the structural changes which occur during healthy ageing. More recently, a wide range of sophisticated machine learning methods have been employed [7, 8, 9]. [4] employed Gaussian process regression to predict the biological age of subjects using structural neuroimaging data, demonstrating that such a model was able to accurately predict brain age. The resulting model was subsequently applied to subjects with traumatic brain injury (TBI), where the associated residuals (difference between predicted and true biological age) were shown to be significantly larger for subjects with TBI as compared with healthy subjects; the associated model consistently predicted subjects with TBI to be *older*, possibly a result of accelerated atrophy. This work was further extended by [10], who employed convolutional neural networks to obtain improved performance. In related work, [11] employ kernel regression with an application to the early identification of Alzheimer's disease.

While the vast majority of the literature has employed structural imaging modalities, there are also numerous examples of where functional imaging has been utilized. A pertinent example is [12], who employ resting-state fMRI together with support vector machines (SVMs) in order to accurately classify subjects as being either children (ages 7-11 years old) or adults (ages 24-30 years old). Furthermore, they observe an overall decrease in network connectivity as subjects mature. In related work, [13] identify ageing-driven changes in functional connectivity, highlighting decreased connectivity within the default mode network and the somato-motor network. Subsequently, [14] categorized the changes in functional connectivity that occur with healthy ageing in terms of various network measures.

More generally, the study of functional connectivity is itself an exciting avenue of modern neuroscientific research which has shown great potential for improving our understanding of the human brain function and architecture [15]. By way of example, changes in functional connectivity have been related to various neuropathologies such as Parkinson's disease [16] and Alzheimer's [17] as well as conditions such as Autism [18]. Recently, the changes in functional connectivity induced by ageing have begun to be studied. Initial studies have reported significant differences in the connectivity between younger and older subjects using resting-state fMRI [14]. Moreover, results appear to suggest there are important changes that occur in the connectivity not just between regions but also at the level of entire networks. However, despite recent advances, a holistic understanding of the relationship between healthy ageing and the associated changes in functional connectivity is still missing.

In this work we seek to build robust models of brain age based on the functional connectivity of individuals. This serves to combine the two prominent avenues of neuroscientific research: brain age prediction and analysis of functional connectivity. In particular, the methods presented in this work have two principal objectives:

1. To demonstrate that measures of functional connectivity can reliably be employed as features in machine learning models of brain age. To this end we build and validate models using three large open-source datasets: the Cambridge Center for Ageing and Neuroscience (CamCAN), the Human Connectome Project (HCP) and the ATR Wide-Age-Range datasets.

2. We further wish to interpret and inspect the proposed models in order to gain further insights into the changes in functional connectivity associated with ageing. This calls for the

use of parsimonious and simple predictive models together with features whose relationship with functional conncetivity is clearly understood.

Throughout this paper, we put forward the thesis that for the potential impact of functional connectivity assessment to be met (i.e., in terms of developing powerful biomarkers) the research community needs to develop robust methods for data-analysis which can combine both supervised and unsupervised models of functional connectivity analysis. Instead of tweaking existing statistical methods, it is imperative to develop methods which are intuitive, interpretable, and insightful from a neurophysiological perspective. Such models must utilise as much experimental information as possible in order to investigate the factors which affect functional connectivity.

To further motivate our thesis, one should consider that most experiments to date operate on data from a single laboratory, or class of experiment which limits the generality of any obtained results. Such concerns have been recently recognised, particularly within the context of brain ageing [19, 20], and have given rise to multi-laboratory collaborations with data-sharing becoming more common. However, it is still highly unlikely that all subject features (and how these are measured) will be comparable across different experimental environments. Thus while data-sharing has seen much progress, it could be argued that the impact of these endeavours is still to come, and to achieve this, we need to develop methods which can combine information from across disparate, but informative experiments.

To this end we proceed in a two-step framework. First, we seek to learn robust features which summarize properties of functional connectivity across a cohort of subjects in an unsupervised manner. Due to our focus on interpretability, we focus on linear latent variable models, such as principal component analysis (PCA), independent component analysis (ICA) and their generalizations. The benefit of employing latent variable models such as PCA is that we may interpret the latent variables in terms of activity within functional connectivity networks, as proposed by [21] (see also Fig 2 below). Second, once features have been obtained in an unsupervised manner, they are subsequently used to predict brain age using standard linear regression models. We deliberately restrict ourselves to simple linear classifiers as they can be easily interrogated, allowing us to explicitly understand how each feature contributes to the predicted brain age. An overview of our two-stage approach is provided in Fig 1.

The remainder of this manuscript is organized as follows: in Section 2 we first review linear latent variable models and their implications for functional connectivity analysis. We then present our proposed two-step procedure. Experimental results, studying synthetic as well as real resting-state fMRI data, are presented in Section 3.

## 2 Materials and methods

We focus our analysis on resting-state fMRI time series data which is collected across a cohort of $N$ subjects. For the $i$th subject, it is assumed we have access to fMRI measurements over $p$ fixed regions of interest, denoted by $X^{(i)} \in \mathbb{R}^p$, as well as the subjects age, $a^{(i)} \in \mathbb{R}_+$. Throughout this work we approximately model the fMRI data for each subject with a stationary multivariate Gaussian distribution, $X^{(i)} \sim \mathcal{N}(0, \Sigma^{(i)})$, where $\Sigma^{(i)}$ denotes the covariance for subject $i$. Each entry in $\Sigma^{(i)}$ denotes the covariance between any pair of regions, which serves to define a measure of the functional connectivity [22]. As such, it follows that $\Sigma^{(i)}$ encodes a functional connectivity network over $p$ regions where edges encode the marginal dependence structure.

The goal of the proposed methods is to learn interpretable and robust models to predict the biological age, $a^{(i)}$, of subjects given information relating only to their functional connectivity. To achieve this, we propose a two-step framework. Our approach first employs linear latent variable models in order to model high-dimensional connectivity matrices using a reduced

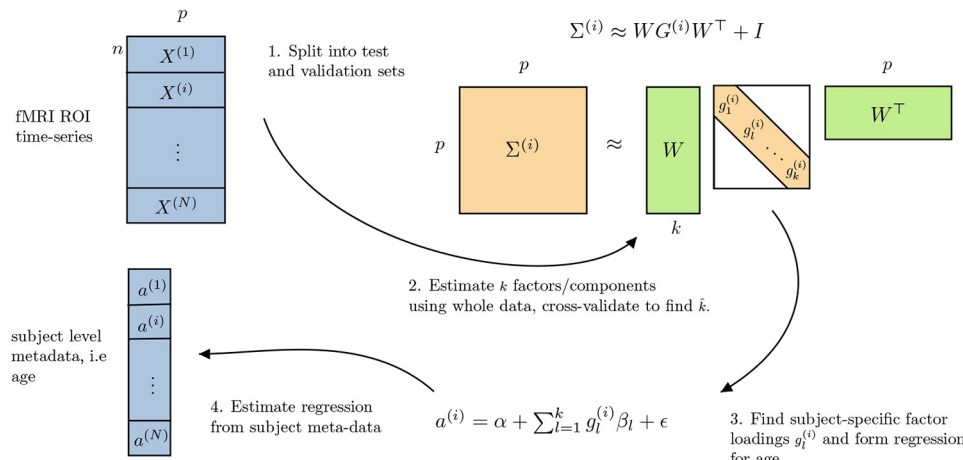

**Fig 1. Pipeline for estimating networks, factor loadings, and predictive model for biological brain age.** Inferred factors $W \in \mathbb{R}^{p \times k}$ describe networks which are reproducible across the entire population, the subject-specific factor loadings $g_l^{(i)}$ are then used to predict brain age. Once the factor loadings are estimated as above, using one experimental data-set (we use CamCAN data in our experiments), we can then assess how these factors perform for brain age prediction on completely held-out data-sets; we demonstrate how the model generalizes well using HCP and ATR Wide-Age-Range datasets.

number of latent variables. We interpret such variables as corresponding to functional connectivity networks, allowing us to describe patterns in connectivity as being composed of various distinct networks. We note that such a two-step approach has previously been employed in the context of brain age prediction [11, 9]. However, as far as we are aware, this is the first work to directly interpret the role of linear latent variable models, such as PCA, as learning the relevant functional networks. This work thereby provides a clear motivation and interpretation for such a two-stage strategy.

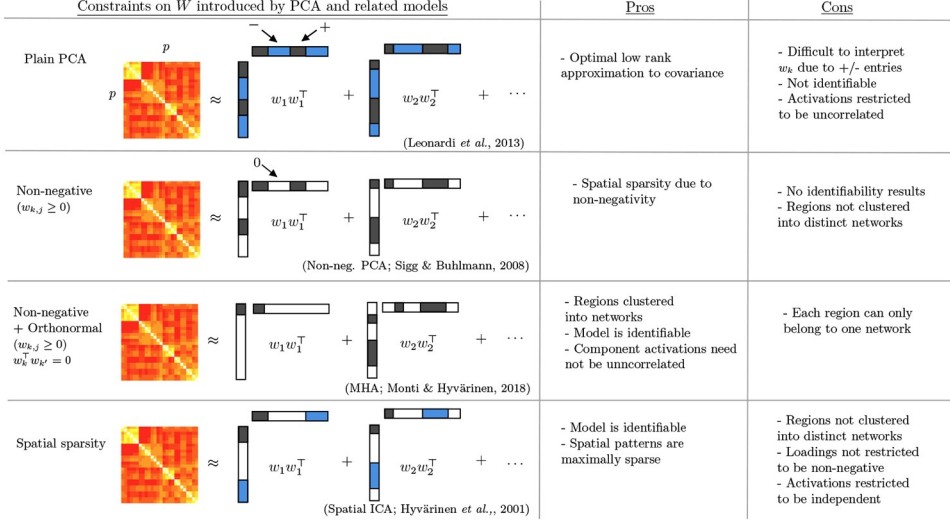

**Fig 2. Figure demonstrating the relationship between linear latent variable models, such as PCA and its extensions, to inferred networks.** We highlight how introducing various structural constraints on the loading matrix, $W$, improves interpretability of such models.

In Section 2.1 we discuss the various latent variable models employed, and highlight how introducing assumptions such as non-negativity can help further improve interpretability of results. We also discuss theoretical benefits associated with such assumptions. We then discuss the how the features (i.e., functional networks) inferred by the latent variable models may be used to build linear models for brain age.

## 2.1 Linear latent variable models for functional connectivity: PCA and its extensions

In this section we outline the linear latent variable models employed in the unsupervised learning stage of the proposed framework. We begin by discussing principal component analysis (PCA), a well-established technique for dimensionality reduction [23]. The common derivation for PCA poses it as an optimization problem seeking to learn the linear projection which maximizes explained variance within the projected space [24]. However, PCA can also be derived as inference under a simple linear latent variable model, which posits that observations $X^{(i)} \in \mathbb{R}^p$ are generated as a linear projection from low-dimensional latent variables, $Z^{(i)} \in \mathbb{R}^k$ [25]. When both observations and latent variables are taken to follow a multivariate Gaussian distributions we obtain the following generative model for observed data:

$$Z^{(i)} \sim \mathcal{N}(0, G^{(i)}) \tag{1}$$

$$X^{(i)}|Z^{(i)} = z^{(i)} \sim \mathcal{N}(Wz^{(i)}, v^{(i)}I) \tag{2}$$

where $G^{(i)} \in \mathbb{R}^{k \times k}$ is a diagonal matrix and $v^{(i)} \in \mathbb{R}_+$ denotes measurement noise. Eqs (1) and (2) serve to highlight how PCA can be seen as a low-rank model for the covariance matrix; by marginalizing over latent variables we obtain:

$$\Sigma^{(i)} = WG^{(i)}W^T + v^{(i)}I, \tag{3}$$

implying that the loading matrix, $W$, captures low-rank covariance structure. Learning the associated loading matrix, $W$, proceeds via maximizing the log-likelihood over observations across all $N$ subjects:

$$\mathcal{L} = \sum_{i=1}^{N} p \log 2\pi + \log \det \Sigma^{(i)} + \text{tr}(\Sigma^{(i)^{-1}} K^{(i)}), \tag{4}$$

where $\Sigma^{(i)}$ is as defined in Eq (3) and $K^{(i)}$ denotes the sample covariance matrix for the $i$th subject. In the context of PCA, the maximization is performed subject to the constraint that $W$ be orthonormal,

$$\hat{W} = \underset{W:W^TW=I}{\arg \max} \{\mathcal{L}\}, \tag{5}$$

and a closed-form solution is obtained via eigendecomposition.

Following [21] it is possible to interpret each column of $W$ as encoding functional networks or "eigenconnectivities". While the loading matrix, $W$, is shared across all subjects, each diagonal entry of $G^{(i)}$ denotes the extent to which the associated network is expressed in subject $i$. This allows us to study connectivity as being composed of various distinct networks, resulting in significant benefits from the perspective of interpretability. We can further unpack Eq (3) as

follows (see also Fig 2 below):

$$\Sigma^{(i)} = \sum_{j=1}^{k} g_j^{(i)} W_j W_j^T + \nu^{(i)} I, \qquad (6)$$

where $W_j$ denotes the $j$th column of $W$ and we write $g_j^{(i)}$ to denote the $j$th diagonal entry of the matrix $G^{(i)} \in \mathbb{R}^{k \times k}$. As such, we may interpret each $W_j$ as encoding the $j$th network and $g_j^{(i)}$ as a measure of activity within the corresponding network in the $i$th subject.

There exist several extensions to the model described in Eqs (1) and (2), the prime example being factor analysis which allows the variances in Eq (2) to vary across dimensions. Recently, several extensions have been proposed where constraints such as non-negativity are introduced with the goal of improving the interpretability of results [26, 27, 28]. The motivation behind such methods stems from the fact that interpreting and visualizing PCA-based networks becomes very challenging, particularly in high-dimensions. Challenges arise from the fact that each principal component will correspond to a weighted sum of BOLD activities across all observed regions. As such, it is often difficult to identify which regions are the principal contributors to a certain principal component (and hence functional network) without applying ad-hoc post analysis. Furthermore, it is possible that some entries in the principal components may be negative, which further complicates the interpretation from the perspective of functional connectivity analysis.

The aforementioned issues can be mitigated via the introduction of non-negativity constraints on the loading matrix, $W$. This ensures that each principal component corresponds only to a weighted *positive* sum of activity over all brain regions. As such, the principal component can be directly interpreted as the contribution of each region to each functional network. Furthermore, the introduction of non-negativity will often yield sparsity in the sense that many of the entries of the principal components will be exactly zero [27]. It follows that such sparsity further facilitates the interpretation of the corresponding networks. From an optimization perspective, the loading matrix is inferred by maximizing the original log-likelihood objective, with the additional non-negativity constraint:

$$\hat{W} = \underset{W:W\geq 0}{\arg\max}\{\mathcal{L}\}. \qquad (7)$$

It is important to note that the orthonormality constraint has been dropped in Eq (7), making the associated optimization problem less challenging. However, the combination of non-negativity and orthonormality, as enforced in [29], leads to several desirable properties. First, the loading matrix $W$ has at most one non-zero entry per row. This implies that we may interpret the columns of $W$ as encoding membership to $k$ non-overlapping networks or clusters. Another very important benefit of introducing non-negativity and orthonormality constraints is that the matrix $W$ is uniquely defined and identifiable. This is not the case in standard factor analytic models, where $W$ is only identifiable up to an arbitrary rotation [30, 25]. Given that throughout this work we will directly interpret the columns of the loading matrix, $W$, as encoding functional connectivity networks, the lack of identifiability in PCA and factor analysis models is a significant limitation. We refer to the model presented in [29] as Modular Hierarchical Analysis (MHA). The associated optimization problem therefore becomes:

$$\hat{W} = \underset{W:W^T W=I \text{ and } W\geq 0}{\arg\max}\{\mathcal{L}\}. \qquad (8)$$

MHA can therefore been seen to address the two important limitations of traditional models such as PCA and factor analysis; first that the presence of negative values in the loading

matrix complicates the interpretation of such matrices (addressed via the use of non-negativity constraints) and second is the fact that the latent variables are rotationally invariant (addressed via the further introduction of orthogonality). A further limitation of models such as PCA and factor analysis is that they implicitly assume latent variables must be uncorrelated. In many cases, especially when such models are applied on data relating to a cohort of subjects, such an assumption will not be valid, implying the associated generated models are misspecified. In contrast, MHA is able to identify and recover components even when they are uncorrelated. This is an important theoretical advantage, as MHA continues to enjoy the same identifiability properties even in the presence of correlated latent variables, and practical advantage, as we demonstrate in this work. Finally, we note that in the context of fMRI data, MHA corresponds to an intuitive generative model whereby latent variables capture the activity within each functional network. The optimization of Eqs (5), (7) and (8) is discussed in S1 Appendix. Furthermore, we provide both Python and R code to implement MHA in S1 Code.

Moreover, we note that model introduced by [26], termed Modular Connectivity Factorization (MCF), shares many similarities with MHA. In fact, both methods introduce non-negativity and orthonormality over the loading matrix, $W$. The fundamental difference, however, is that MCF is not associated with a linear latent variable model, and instead parameters are inferred as follows:

$$\hat{W} = \underset{W:W^T W=I \text{ and } W \geq 0}{\arg\max} \left\{ \sum_{i=1}^{N} \text{tr}(\Sigma^{(i)} K^{(i)})^2 \right\}, \tag{9}$$

where $\Sigma^{(i)}$ is defined as in Eq (6) and $K^{(i)}$ is the empirical covariance for the $i$th subject. A related approach was also proposed by [31].

Finally, it is important to note that whilst identifiability can be obtained via the combination of non-negativity and orthonormality, as is the case with the MHA model, it can also be obtained by relaxing the assumed distribution over latent variables, as is the case with independent component analysis (ICA) models. Formally, ICA is also a linear latent variable model, however, latent variables are no longer assumed to follow a Gaussian distribution [32]. While the relaxation of the Gaussianity assumption complicates the associated optimization, which must now be solved using gradient descent methods and accounting for the presence of multiple local optima due to the non-convex objective function [33], ICA has been widely employed in the study of functional connectivity [34, 35]. Moreover, we note that the "spatial" version of ICA used in fMRI reverses the roles of latent variables and loadings, which means that it is actually looking at the non-Gaussianity or sparsity of what we call here the loadings, corresponding to spatial patterns. Fig 2 provides a visualization of the benefits obtained by introducing each of the aforementioned constraints. In particular, we note that it is the combination of non-negativity together with orthonormality which yields interpretable and identifiable networks. We empirically validate such claims by applying all of the aforementioned models to synthetic and real fMRI datasets below.

## 2.2 Predicting brain age using functional network activity

The previous section outlined the various flavours of latent variable models which can be employed in order to learn functional networks across a cohort of $N$ subjects. The aforementioned models allow us to decompose observed functional connectivity patterns as a linear sum of networks encoded by the columns of the loading matrix, $W$. While the loading matrix is shared across all subjects (indicating the same networks are present across all subjects), the extent to which they contribute to the observed covariance of the $i$th subject is denoted by the diagonal entries of $G^{(i)}$, as stated in Eq (6).

We now consider the task of predicting the biological brain age, $a^{(i)}$, using inferred functional connectivity networks as features. In the interest of interpretability we limit ourselves to linear regression models of the form:

$$a^{(i)} = \sum_{j=1}^{k} \beta_j g_j^{(i)} + \epsilon^{(i)}. \tag{10}$$

Recall that $g_j^{(i)}$ corresponds to the $j$th diagonal entry of the matrix $G^{(i)}$. As such, the proposed models will essentially seek to predict the biological age of subjects by considering activity within each inferred functional network. In the case of the $i$th subject, the observed activity in network $j$ is quantified by $g_j^{(i)} \in \mathbb{R}_+$. In practice, we will seek to quantify the activity of various functional networks on unseen subjects, defined to be subjects whose data was not employed to estimate loading matrix, $W$. We note that due to the orthonormality of $W$, together with Eq (6), we may estimate $g_j^{(i)}$ for data from unseen subjects, denoted by $i^*$, as follows:

$$\hat{g}_j^{(i^*)} = W_j^T \hat{\Sigma}^{(i^*)} W_j - \nu^{(i^*)}. \tag{11}$$

We note that Eq (11) requires the observation noise, $\nu^{(i^*)}$. This is not a concern for all subjects whose data is employed during the unsupervised learning of the latent variables, as parameters $\nu^{(i)}$ are inferred alongside loading matrix, $W$. However, the primary goal of this work is to build predictive models which can generalize to unseen subjects. In this context, an estimate of the observation noise, $\nu^{(i^*)}$, can be obtained as follows:

$$\hat{\nu}^{(i^*)} = \text{tr } \hat{\Sigma}^{(i^*)} - W^T \hat{\Sigma}^{(i^*)} W. \tag{12}$$

Although the class of models considered in Eq (10) may be considered amongst the simplest supervised regression models, they yield several important benefits when seeking to understand both the estimated parameters as well as the contribution of each of the features. In particular, each $\beta_j$ corresponds to the regression coefficient summarizing the (linear) relationship between the activity of the $j$th network and biological age, conditional on all remaining networks. As such, if certain regression coefficients are deemed to be insignificant, we may conclude that the associated network is invariant during healthy ageing.

## 2.3 Hyper-parameter selection

The proposed two-stage estimation framework requires the input of only one hyper-parameter: the dimensionality of latent variables $k$. In the context of PCA and factor analysis, this hyper-parameter directly corresponds to the number of principal components or factors inferred, and a wide literature exists for tuning such a parameter [23]. One of the advantages of the latent variable models presented in Section 2.1 is that they each correspond to probabilistic models whose likelihood can be directly evaluated. As such, a logical choice to tuning hyper-parameter $k$ is to directly maximize the log-likelihood over held out data.

In order to effectively perform hyper-parameter tuning as well as quantify the generalization performance of the proposed method, data was split into training, validation and test datasets as follows:

- First, a subset of subjects were held out as test data. As such, we obtain two datasets:

$$\left\{ X_{1:n}^{(i)}, a^{(i)} \right\}_{i \in S_{train}} \quad \text{and} \quad \left\{ X_{1:n}^{(i)}, a^{(i)} \right\}_{i \in S_{test}}$$

where $S_{train}$, $S_{test} \subset \{1, \ldots, N\}$ denote the non-overlapping sets of training and test subjects

respectively. Recall $N$ is the number of subjects present and we write $X_{1:n}^{(i)}$ to denote the $n$ observations available for the $i$th subject.

- Training data is further split into training and validation datasets on a subject-by-subject basis.

  Splitting the data in this manner allows for effective hyper-parameter tuning, using training and validation datasets, as well as for generalization performance to be measured using test dataset which corresponds to unseen subjects.

## 2.4 Experimental data

The data employed in this manuscript corresponds to resting-state fMRI data taken from three distinct open-access repositories. There were small variations in the resting-state functional MR image acquisition for each of the repositories considered: CamCAN [38], Human Connectome Project [37], and the ATR Wide Age Range [38]. The pre-processing employed on each dataset was as follows:

- CamCAN: This dataset was pre-processed by us. Data was motion corrected, spatially smoothed with a 5mm FWHM Gaussian kernel, registered into MNI152 standard space using FLIRT [39] via a skull-stripped high-resolution T1 image and resampled to 4x4x4mm voxel sizes. Each high resolution T1 image was segmented into grey and white matter and cerebrospinal fluid using SPM Dartel [40]. Mean timecourses for cerebrospinal fluid and white matter as well as 6 motion parameters were linearly filtered from each voxel to reduce non-neural noise.

- HCP: We used the pre-processed resting-state fMRI data from a random subset of healthy participants. Notably, the pipeline involved FIX ICA-based noise reduction process [40], to remove individual sources of physiological, non-physiological and motion related noise. Full details of the pre-processing pipeline can be found at https://www.humanconnectome.org/study/hcp-young-adult/document/extensively-processed-fmri-data-documentation.

- ATR: We used the preprocessed data. The pre-processing pipeline notably included regressing out the global grey matter signal as well as signals from cerebrospinal fluid and white matter, to remove sources of spurious variation

  All three pre-processed fMRI datasets were subsequently processed as follows: a cortical parcellation based on resting-state functional connectivity analyses [42] was used to define 264 distinct 10mm diameter regions of interest (ROIs). The fMRI time course averaging across all voxels within each ROI was extracted. These 264 average time courses were then used in subsequent analyses. Full details are provided here https://bicr-resource.atr.jp/var/www/webapp/bicrresource/bicrresource/staticfiles/pdf/Methods.pdf.

## 3 Results

In this section we present a range of experimental results involving both synthetic and real resting-state fMRI datasets. Throughout this section, we contrast the performance of the various linear latent variable models presented in Section 2.1. In particular, we study the performance across the following methods: factor analysis (FA), PCA, non-negative PCA [27], MCF [26] and MHA [29] as well as ICA. In the case of ICA, we first employ PCA as a dimensionality reduction before employing the FastICA algorithm proposed by [43]. The implementations available in `Scikit Learn` were employed for Factor Analysis, PCA and ICA [44].

We first present results using synthetic data in Section 3.1. These simulation experiments serve as a numerical validation of the proposed two-stage procedure. Experiments relating to brain age prediction from resting-state fMRI data are subsequently presented in Section 3.2.

## 3.1 Synthetic data experiments

In this section we evaluate the performance of the proposed two-stage estimation framework using synthetic data. To this end, we generate artificial data whose properties approximately match those which are frequently reported in fMRI studies. The objective is then to quantify which of the linear latent variable models presented in Section 2.1 are able to both robustly recover the associated loading matrix, $W$, as well as learn the relevant factors which serve as accurate predictors of brain age on unseen subjects. Synthetic data was then generated in order to satisfy Eqs (1), (2) and (10). This is achieved as follows:

- First, we randomly generated a factor loading matrix, $W \in \mathbb{R}^{p \times k}$, which satisfied the constraints of both non-negativity and orthonormality. The reason for introducing both constraints is that we will seek to quantify how reliably each latent variable model can recover $W$, and it is therefore imperative to ensure we generate $W$ from an identifiable model (see discussion in Section 2.1). In order to achieve this a dense matrix, $W$, was sampled with each entry following a uniform distribution over the interval [0, 1]. Subsequently, for each row only the entry with the largest value was retained with all other entries set to zero. Finally, the norm of each column was set to one.

- Second, the factor loadings for the $i$th subject, $g^{(i)} \in \mathbb{R}^k$, were randomly generated as follows:

$$g_j^{(i)} \sim \mathcal{N}(2.5, 1.0), \quad \text{for } j = 1, \ldots, k$$

  with all negative samples being discarded.

- The regression coefficients, $\beta \in \mathbb{R}^k$, were drawn uniformly at random from the interval [0, 10].

- Finally, we are able to randomly generate observations and ages for each subject as follows:

$$X^{(i)} \sim \mathcal{N}(0, WG^{(i)}W^T + \nu^{(i)}), \tag{13}$$

$$a^{(i)} \sim \mathcal{N}(\beta^T g^{(i)}, \epsilon). \tag{14}$$

Recall that $G^{(i)} \in \mathbb{R}^{k \times k}$ is a diagonal matrix consisting of entries $g_j^{(i)}$.

We note that the choices for sampling distributions of both the factor loadings, $g^{(i)}$, as well as the regression coefficients, $\beta$, are necessarily somewhat heuristic. However, care was taken to ensure the implied distributions over subject ages approximately matched the empirical distributions observed within the CamCAN repository.

We note that throughout experiments we consider the performance of each method whilst varying two distinct factors: the number of observations per subject, $n$, and the number of training subjects, $N$. Furthermore, throughout simulations we fix the dimensionality of observations to be $p = 50$ and the number latent factors to be $k = 5$.

Given artificial data generated as described above, we look to quantify the performance of each of the linear latent variable models using the following two metrics:

1. Accurate recovery of the loading matrix, $W$. This is quantified in terms of the squared error between the true loading matrix and the estimated loading matrix.

2. Accurate brain age prediction over unseen subjects. In line with other literature, this is quantified in terms of the mean absolute error between true and predicted brain ages [11, 8].

**3.1.1 Synthetic data results.** We begin by considering the performance of each linear latent variable model as the number of observations per subject, $n$, increases for a fixed number of training subjects, $N = 25$. The results are presented in Fig 3. We note that both in terms of recovery of the loading matrix, $W$, as well as in terms predicting the ages over unseen subjects, the introduction of regularity constraints, be they in the form of non-negativity, orthonormality or non-Gaussianity or sparsity (as in ICA), leads to improvements.

We also study the performance of the various latent variable models when the number of training subjects, $N$, increases and the number of observations is fixed at $n = 100$ per subject. These results are presented in Fig 4. In terms of recovery of the loading matrix, $W$, we again observe that introducing regularity constraints leads to significant improvements. In terms of

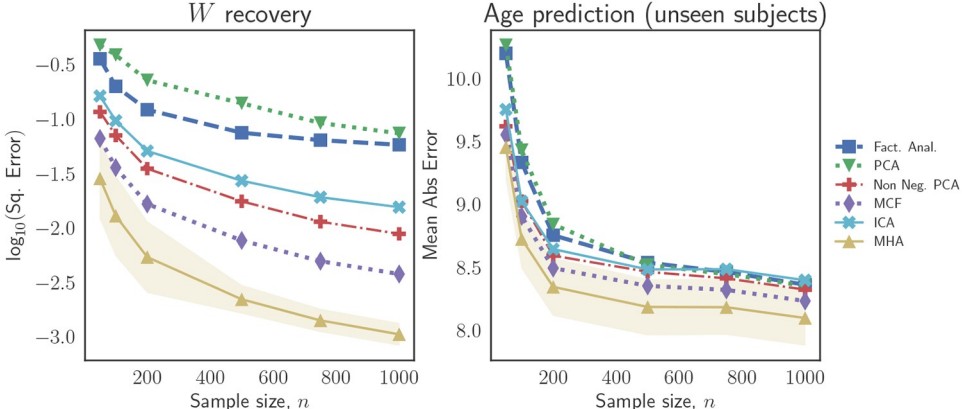

**Fig 3.** Simulation results for recovery of the true loading matrix (left panel) and prediction of brain age for unseen subjects (right panel) as the number of observations per subject, $n$, increases. We note that the introduction of regularity constraints (e.g., non-negativity or orthonormality) on the loading matrix leads to improvement in performance.

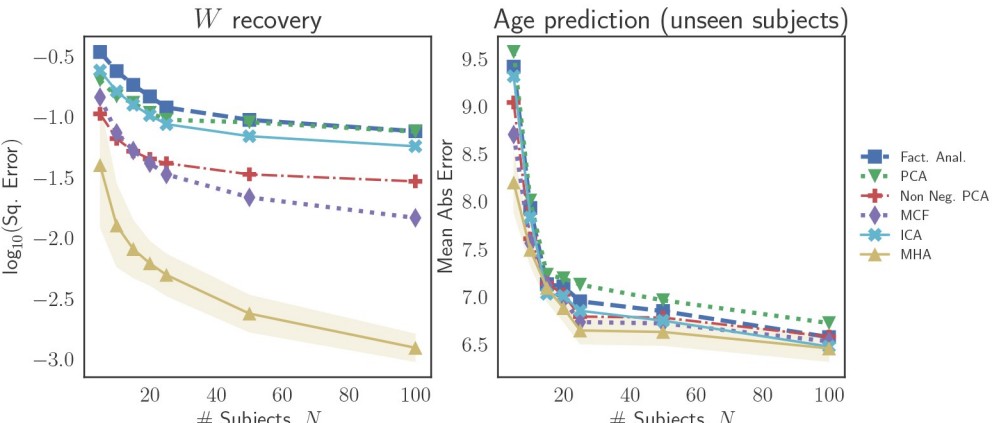

**Fig 4.** Simulation results for recovery of the true loading matrix (left panel) and prediction of brain age for unseen subjects (right panel) as the number of training subjects, $N$, increases. We note that the introduction of regularity constraints (e.g., non-negativity or orthonormality) on the loading matrix leads to improvement in performance.

predictions over unseen subjects (as shown in the right panel of Fig 4), the improvements due to the introduction of regularity conditions begin to fade as the number of training subjects increases. In particular, beyond a certain number of training subjects (approximately 25 in the case of these experiments), the improvement in out-of-sample predictions begins to plateau.

## 3.2 Resting-state fMRI data experiments

While the previous section presented results relating to synthetic data, here we present experimental results where the proposed two-step procedure is applied to three open-source resting-state fMRI datasets. The datasets considered correspond to the Cambridge Center for Ageing and Neuroscience (CamCAN) repository, the Human Connectome Project (HCP) repository, and the ATR Wide-Age-Range repository. The purpose of employing three distinct datasets is to effectively measure the generalization performance of the proposed approach on unseen data. As such, data from the HCP and Wide-Age-Range repositories was not employed during any of the model training and instead used exclusively as unseen test data. It is important to note that in addition to significant inter-subject variability [45], fMRI data also suffers from the presence of several other well-documented issues such as variable scanner performance or noise [46, 47, 48]. As such, validating the performance of the proposed brain age prediction models in this way will provide a more realistic measure of their generalization performance.

**3.2.1 CamCAN repository results.** Resting-state fMRI data was collected from a total of 647 subjects from the CamCAN repository. Subject ages ranged from 18 to 88 years of age (average age of 54.31±18.56, 318 males and 329 females). The CamCAN dataset was employed as the principal dataset in the proposed two-step procedure, implying that it was employed to learn both the functional network structure in the unsupervised learning stage and the linear regression models in the supervised learning stage. As such, the data was split into training, validation and test subsets as described in Section 2.3.

*Step 1*: *Unsupervised functional network inference*. The first stage of the proposed framework involves the estimation of reproducible functional connectivity networks via the use of the various linear latent variable models discussed in Section 2.1. The number of functional networks inferred corresponds directly to the dimensionality of latent variables, which is determined by hyper-parameter $k$. As each linear latent variable model can be interpreted as a probabilistic model, we select hyper-parameter $k$ by maximizing the log-likelihood over the validation dataset. This resulted in the choice of $k = 5$ when the loading matrix was restricted to be both non-negative and orthonormal, as proposed by [26] and [29]. While it is possible that the choice of hyper-parameter may vary across distinct latent variable models (e.g., for PCA or factor analysis), we choose to keep the choice of $k$ fixed across all models as this facilitates model comparison and interpretation of results.

The left panel of Fig 5 visualizes the results when the MHA linear latent variable model was employed (Figures produced using the `plot glass brain` function from the `nilearn` python module [49]). We note that, as discussed in Section 2.1, the MHA linear latent variable model effectively clusters regions into sub-networks via the introduction of non-negativity and orthonormality constraints. As such, each plot in the left panel of Fig 5 visualizes spatially remote brain regions which have been clustered together, indicating that these regions share strong positive correlations. We note that these correlations (i.e., edges in a network) are omitted for clarity in Fig 5. The results demonstrate that the inferred networks are spatially homogeneous and symmetric across both hemispheres. Furthermore, many of the inferred networks correspond to widely reported networks and regions: network 1 captures the default model network (DMN) and network 2 overlaps with the salience network, while networks 3 and 4 correspond to a higher-level visual network and the somatomotor network respectively. For

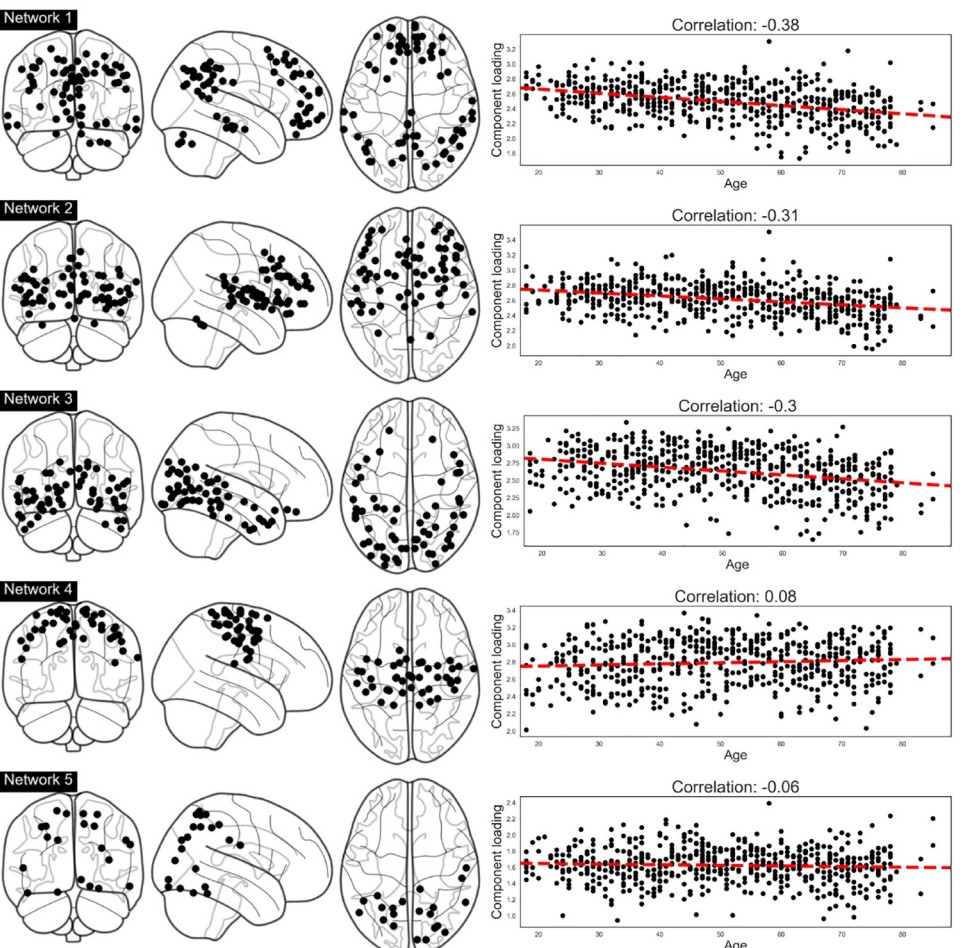

**Fig 5.** Left panel: Inferred networks as recovered when non-negativity and orthonormality constraints are introduced over the loading matrix, *W*. Networks are spatially consistent and symmetric. Right panel: visualization of network activities against subject age demonstrating (mostly negative) linear trends with healthy aging.

comparison, we include equivalent plots for all other latent variable models considered in visualized in Fig 6, presented in the Supplementary Material. We note that alternative methods, such as PCA, which did not enforce the combination of both non-negativity and orthonormality, yielded results which were visibly less clustered and more difficult to interpret.

The right panel of Fig 5 visualizes the correlation between the activity of each network (as defined in Eq (11)) with the age of each subject. For networks 1-3 we observe a significant negative correlation between the activity and age, suggesting that ageing induces a drop in activity of such networks. These results are in line with related research on ageing induced differences in functional connectivity. In particular, the decrease in activity of the DMN (network 1), has been widely reported [19, 50, 51].

*Step 2*: *Supervised training of brain age prediction models*. Recall that the overall objective of the proposed framework was build interpretable models of biological brain age. To this end, the features recovered from linear latent variable models where employed as features in a linear regression framework to predict the brain age of each subject. In particular, the five distinct the linear latent variable models detailed in Section 2.1 where employed to learn reproducible sub-networks parameterized by a loading matrix, $W \in \mathbb{R}^{p \times k}$. The activity within each

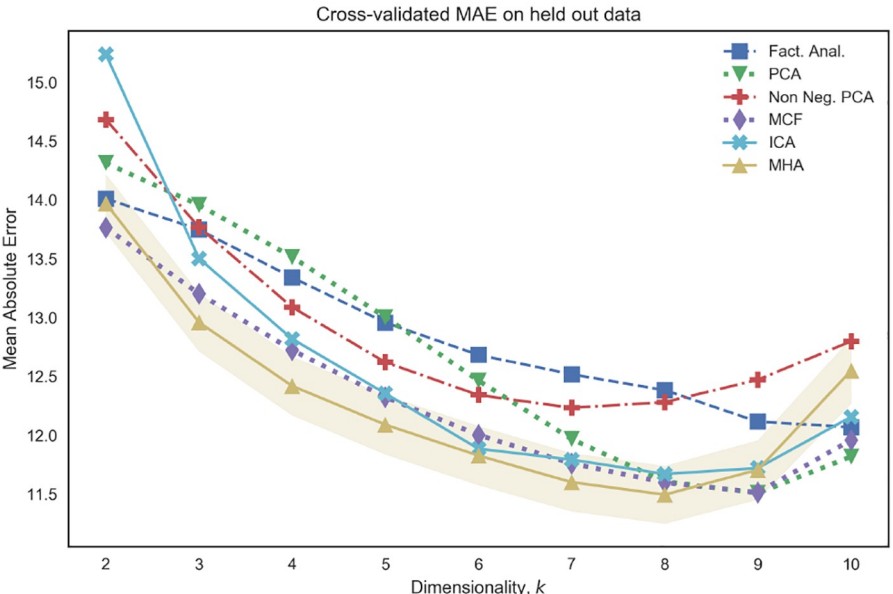

**Fig 6. Inferred networks using alternative linear latent variable models.** In the case of models such as PCA and factor analysis, networks were obtained by thresholding entries of $W$ so only non-negative entries considered.

functional network, defined as in Eq (11), was subsequently employed as features to predict biological age using linear regression.

We note that the CamCAN repository, as well as HCP and ATR repositories, each contained over a hundred subjects each. This is in contrast to typical fMRI studies, where the sample size is often in the range of 20 to 30 subjects [52, 48]. Furthermore, recall that the goal of experiments presented are to quantify performance on unseen resting-state fMRI data with a view to providing an indication of how each of the linear latent variable models employed would perform in a typical fMRI study. As such, throughout the remainder of this section we report the performance, in terms of mean absolute error, over random subsets of 30 subjects from each repository. This corresponds to a form of bootstrapping, where we average results over a random sample of possible *cohorts*. In practice, we report results over 1000 random subsets of 30 subjects for each of the three repositories considered.

Fig 7 visualizes the mean absolute error on unseen test data for various choices of $k \in \{2, \ldots, 10\}$. We note that the combination of linear regression with the use of non-negativity and orthonormality constraints, as advocated by both the MCF and MHA models,

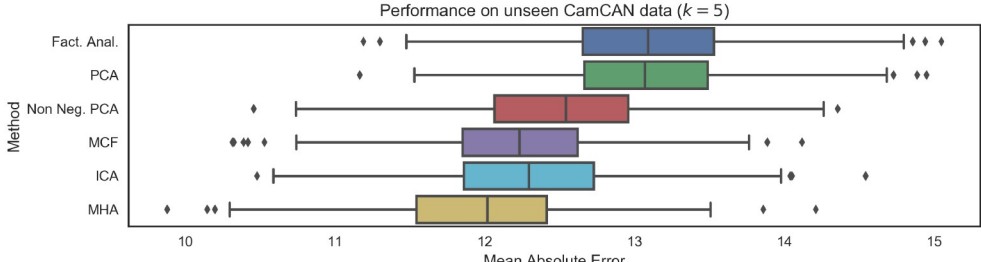

**Fig 7. Mean Absolute Error (MAE) performance for a varying number of networks, as determined by $k$ (x-axis), on unseen test data from CamCAN.** We note that the combination of non-negativity and othonormality (MHA and MCF) yields competitive results across a wide range of $k$.

leads to competitive performance over a range of choices of *k*. In particular, such algorithms out-perform both non-negative PCA and PCA, suggesting that the introduction of such constraints serves to improve the predictive properties of the model. Moreover, we note that Fig 7 indicates the presence of a bias-variance trade-off that is often encountered in supervised learning whereby performance on unseen test data begins to deteriorate as the number of parameters (in our case *k*) increases beyond a certain value.

As mentioned previously, the choice of *k* = 5 was selected in by maximizing log-likelihood over a validation dataset (i.e., in an entirely unsupervised manner—data regarding subject ages was not considered). Fig 8 visualizes the performance on the unseen test dataset for the specific choice of *k* = 5, for all possible choices of linear latent variable models. The results indicate that as additional constraints are introduced to the loading matrix, the generalization capabilities of the models also improve. As such, MCF and MHA, which introduce the most stringent constraints corresponding to *both* non-negativity and orthonormality, obtain the best generalization performance. We also note that ICA is also competitive. Moreover, non-negative PCA, which relaxes the requirement for orthonormality, is the next most competitive latent variable model. Finally, PCA and factor analysis, which relax all the aforementioned constraints, obtain the worst generalization performance.

**3.2.2 Transfer onto HCP and ATR Wide-Age-Range repositories.** The results of Section 3.2.1 provide a measure of performance, in terms mean absolute error in predicted brain age, within a large-scale resting-state fMRI dataset. However, it is widely accepted that in addition subject-specific noise, there are several other significant contributors to noise in fMRI data: these include issues related to scanner noise and frequency of acquisition of images [46, 47, 48]. As a result, in order to thoroughly verify the generalization performance of the proposed methods, we employ resting-state fMRI data from the HCP and ATR Wide-Age-Range repositories. We note that data from the aforementioned repositories was employed only for testing purposes, as such it was not employed to learn the network structure across subjects, nor to tune the parameters of the linear regression models. For a summary of the characteristics of HCP and ATR Wide-Age-Range datasets see Fig 9 and S1 Table in the Supplementary Material.

Prediction of biological age on both the HCP and ATR Wide-Age-Range repositories was performed as follows: First, the loading matrix, $\hat{W}$ was employed to obtain estimated activity within each network, as detailed in Eqs (11) and (12). Subsequently, predictions of biological age were obtained using Eq (10). At each stage both $\hat{W}$ and $\hat{\beta}$ are the parameters inferred using the CamCAN dataset (i.e., there was no fine-tuning of parameters). As a result, performance on both HCP and ATR Wide-Age-Range datasets provide a robust measure of generalization performance to entirely unseen data.

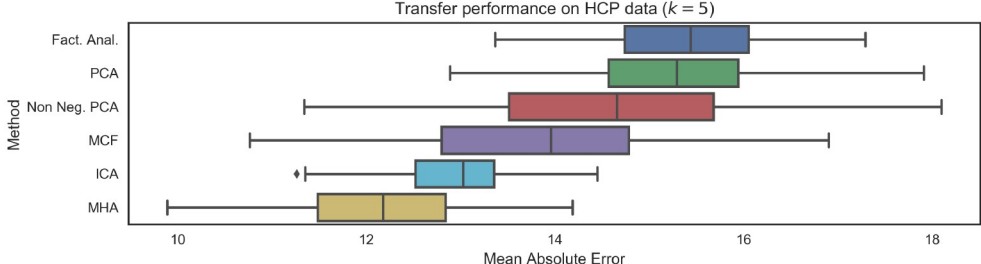

**Fig 8. Mean Absolute Error (MAE) performance on unseen testing data from CamCAN repository when the dimensionality of latent variables is fixed to *k* = 5 (implying we infer 5 networks).** We note that as regularity constraints are introduced, in particular non-negativity and orthonormality, predictive performance improves.

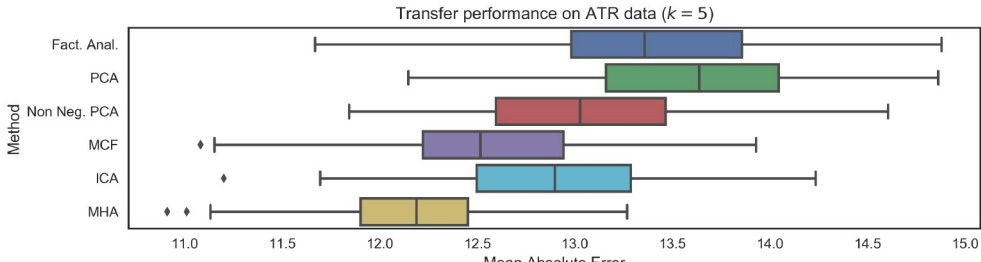

**Fig 9. Histogram visualizing age distribution for each of the repositories employed.** We note that the CamCAN dataset has the widest range of all repositories considered, validating its use as a the primary dataset in our study.

Results on the HCP data are provided in Fig 10. As expected, the mean absolute errors are larger for each of the distinct latent variable models when compared to the results of on the CamCAN dataset (Fig 8), which will be partially the result of varying scanner noise and image acquisition properties. Importantly we note that, as with the CamCAN dataset, there once again a relationship between the introduction of additional constraints (in the form of non-negativity, orthonormality or non-Gaussianity) and generalization performance. As before, methods such as PCA and factor analysis which do not introduce any constraints had the weakest performance as well as the largest drop in performance.

The HCP results presented above serve to partially validate the predictive models trained using the CamCAN dataset. However, one significant limitation of the HCP dataset is that subject ages only range from 22 to 37 years of age. This is particularly relevant in the context of brain age biomarkers, as many neurodegenerative diseases of interest will be associated with advanced ages. As a result, we further validated the generalization capabilities of the proposed

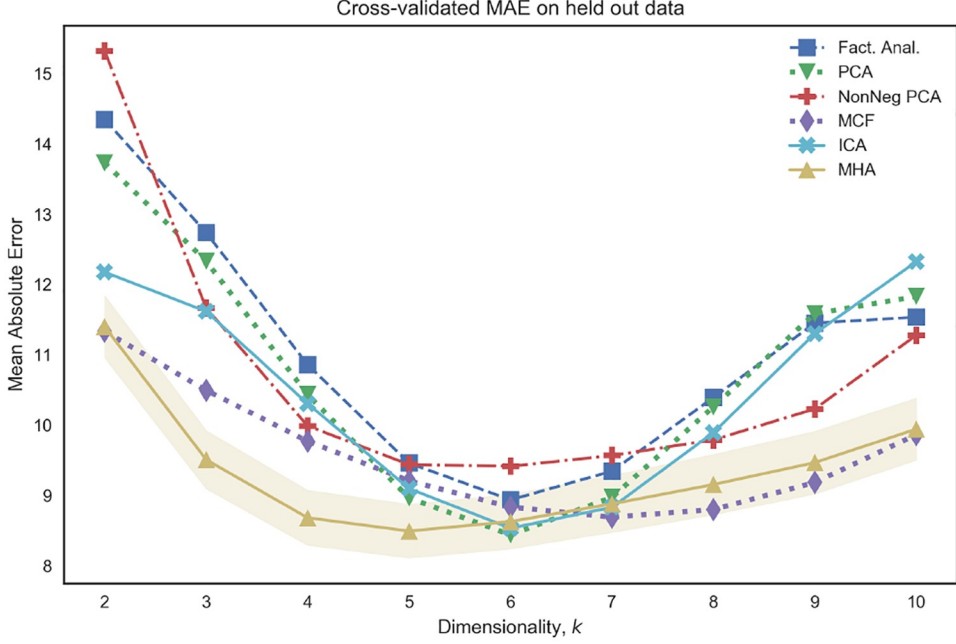

**Fig 10. Mean Absolute Eerror (MAE) performance on unseen data from HCP repository.** Results are broadly consistent with performance on the CamCAN data, indicating good generalization. We note that the introduction of non-negativity or orthogonality constraints leads to improved generalization. The number of functional networks was $k = 5$.

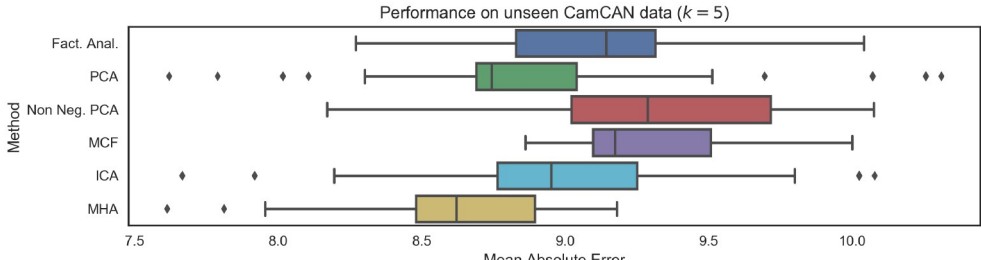

**Fig 11. Mean Absolute Error (MAE) performance on unseen data from ATR Wide-Age-Range repository.** Results are broadly consistent with performance on the CamCAN data, indicating good generalization. Further, as with the HCP data, we note that the introduction of non-negativity or orthogonality constraints leads to improved generalization. The number of functional networks considered was $k = 5$.

brain age prediction models on the ATR Wide-Age-Range dataset, which had subjects ranging from 20 to 70 years of age. Results, presented in Fig 11 are consistent with results on the Cam-CAN and HCP datasets, again indicating that the introduction of constraints non-negativity and orthonormality constraints improves generalization performance.

## 3.3 Extension to non-independent latent variable models

The results presented above employ linear latent variable models where the inferred latents are assumed to be independent. This is clearly stated in the generative model considered in Eq (1) where the covariance of latent variables, $G^{(i)}$, is assumed to be diagonal. Note that in the case of PCA, factor analysis and MHA, since latent variables are assumed to be multivariate Gaussian, the fact the covariance is diagonal implies the latent variables are independent. However, such an assumption will often fail in practice, implying that the empirical covariance structure over latent variables will not be diagonal. In this section we seek to exploit this by directly introducing the off-diagonal entries of the latent variable covariances, $G^{(i)}$, as features in our linear regression models for biological age. As such, whilst Eq (10) considered a linear model where only the diagonal entries of each $G^{(i)}$ were employed to predict biological ages of each subject, we now consider linear regression models of the following form:

$$a^{(i)} = \sum_{j=1}^{k}\sum_{l \geq j} \beta_{jl}\, g_{jl}^{(i)} + \epsilon^{(i)}. \tag{15}$$

Note that in Eq (15) we employ the full upper triangular entries of the covariance matrix as features. This is equivalent to vectorizing the covariance matrix and removing duplicate entries due to symmetry. As such, whilst $k$ features were employed in Eq (10), we now consider a linear models with $\binom{k}{2}$ features; many of which will seek to predict the biological age of individuals based on the off diagonal entries of each $G^{(i)}$. It is important to note that the model presented in Eq (10) is a special case of Eq (15).

As in Section 3.2, we proceed in a two-stage approach whereby we first estimate the loading matrices for the various linear latent variable models employed and subsequently train linear regression models using the full vectorized covariance matrix as features.

Fig 12 visualizes the MAE error on unseen test data as a function of the dimensionality of latent variables, $k$. We note that for all choices of $k$ the reported errors are smaller than those reported in Fig 7. This provides empirical evidence that the off-diagonal entries of the latent variable covariances are discriminative features for brain age prediction, and therefore can be seen as evidence that models which assume diagonal covariance structure over latents are

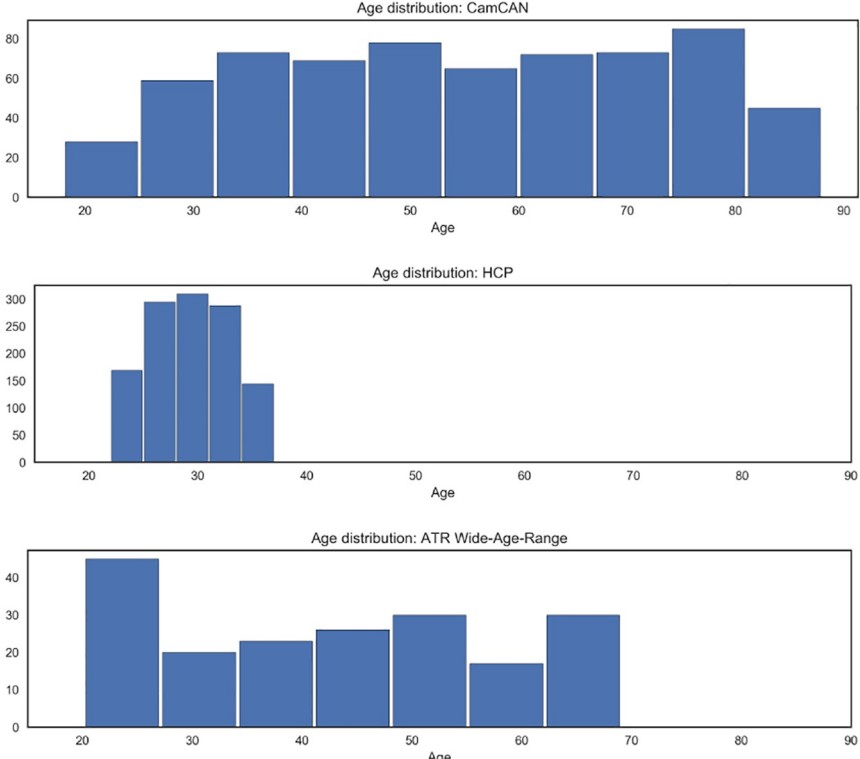

**Fig 12. Mean Absolute Error (MAE) performance for a varying number of networks, as determined by *k* (x-axis), on unseen test data from CamCAN when latent variables are no longer assumed to have an isotropic covariance structure and the full vectorized covariance is employed as features in the linear regression models.** We note that MHA is able to directly accommodate such a scenario and hence is competitive for all choices of latent variable dimension, *k*.

misspecified. Fig 13 provides further visualizations in the case where *k* = 5. We note that the MHA model performs competitively, this is to be expected as this model directly accommodates the possibility of non-independent latent variables [29]. Moreover, we note that MHA performs particularly well when the number of networks is small (when dimension of latent variables, *k*, is less than or equal to 5), which is useful when we wish to prioritize the

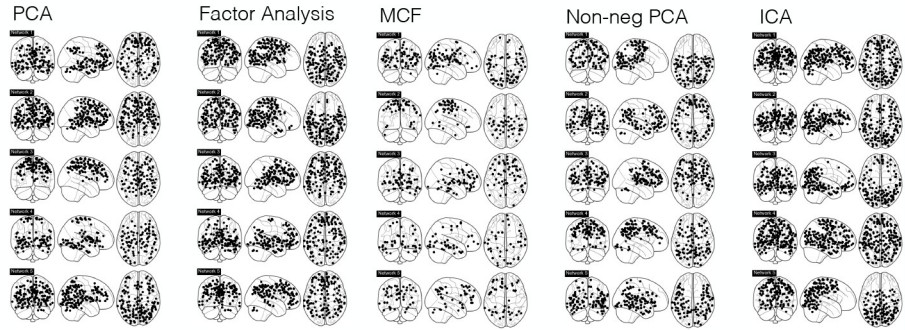

**Fig 13. Mean absolute error (MAE) performance on unseen testing data from CamCAN repository when the dimensionality of latent variables is fixed to *k* = 5 (implying we infer 5 networks).** Note that latent variables are no longer assumed to have an isotropic covariance structure and the full vectorized covariance is employed as features in the linear regression models.

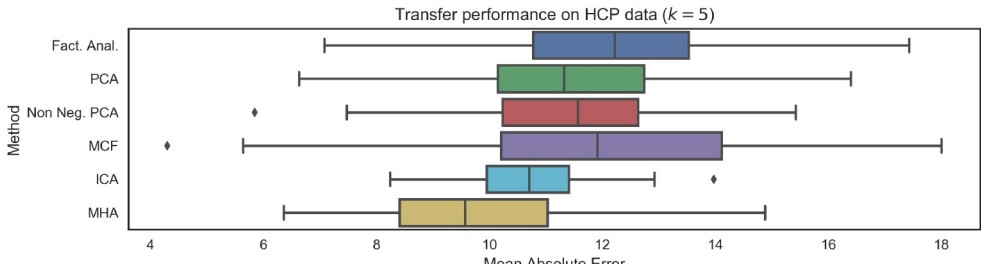

**Fig 14. Mean Absolute Error (MAE) performance on unseen data from HCP repository.** Results are broadly consistent with performance on the CamCAN data, indicating good generalization. We note that the introduction of non-negativity or orthogonality constraints leads to improved generalization. The number of functional networks was $k = 5$.

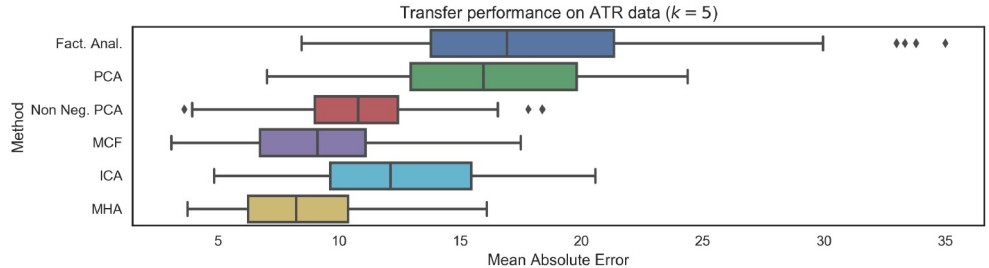

**Fig 15. Mean absolute error (MAE) performance on unseen data from ATR Wide-Age-Range repository.** Results are broadly consistent with performance on the CamCAN data, indicating good generalization. Further, as with the HCP data, we note that the introduction of non-negativity or orthogonality constraints leads to improved generalization. The number of functional networks considered was $k = 5$.

interpretability of results. Finally, the performance of various methods, as depicted in Fig 12, shows similar trends as in Fig 7; there is once again a bias-variance trade-off associated with the choice of $k$ and the introduction of non-negativity or non-Gaussianity constraints (as in MCF or ICA) leads to improved generalization performance. Finally, whilst Fig 12 only shows generalization performance to unseen subjects from the CamCAN cohort, we also present results for generalization performance to brain age prediction on the HCP and ATR Wide-Age-Range datasets in Figs 14 and 15 of the Supplementary Material.

## 4 Conclusion

It is widely accepted that ageing has pronounced effects on the functional architecture of the human brain [14, 9]. In the current study we have presented and validated a two-stage framework through which to train interpretable and robust models of biological brain age based on functional connectivity. In particular, the proposed framework first employs linear latent variable models to uncover reproducible networks which are present throughout a cohort of subjects. A variety of such latent variable models are considered many of which extend PCA by introducing constraints such as non-negativity over the loading matrix. Our experiments suggest that whilst PCA is a natural candidate for dimensionality reduction, and can be interpreted as recovering latent *eigenconnectivities*, the introduction of constraints such as non-negativity can serve to greatly improve both interpretability and predictive performance. While ICA improves on PCA by introducing spatial sparsity, we found that MHA as well as MCF lead to better results, especially in the case of a small number of networks. Reasons for

this improvement include using a combination of non-negativity and orthogonality that leads to disjoint networks, as well as explicit modelling of connectivity between the components.

Given inferred functional networks and their activations we train linear predictive models of biological brain age where in the interest of interpretability we deliberately restrict ourselves to linear models. This allows us to directly interrogate the effects of each functional network on the predicted brain age (as shown in Fig 5). In line with other results in the literature, we find a decrease in activation in the default mode network, salience network and higher-level visual network as biological age increases.

The proposed two-stage framework is first validated on the data from the CamCAN repository and subsequently further applied to two further open-access repositories: the HCP and ATR Wide-Age-Range repositories. The use of data from two additional repositories serves to provide a clear empirical indication of the generalization capabilities of the proposed approach. This is especially relevant in the context of fMRI data, where artefacts such as scanner noise can often cause significant challenges [48].

We note that the brain age prediction errors presented in this work are not competitive with alternative methods which are based on alternative imaging modalities, such as structural imaging data [53, 10]. This is to be expected for two reasons. First, the imaging modality employed in this work, resting-state fMRI data, is both noiser and likely to be less age-indicative than structural measures. Second, in this work we deliberately restrict ourselves to building simple yet interpretable models of brain age. As such, we restrict ourselves to consider only linear classifiers as these allow for clear model interpretation and interrogation, while noting that the use of more expressive models (e.g., nonlinear models) in the second stage should naturally lead to improved performance.

Furthermore, it is important to note that whilst this work demonstrates the feasibility of functional connectivity driven models of biological brain age, all subjects included in these studies were healthy. As such, whilst such models could eventually be employed to develop biomarkers, further experimentation and validation will be required in future. Moreover, an avenue for further research would be to consider performing classification instead of regression in the second stage of the proposed method. Whilst a natural task would be to discriminate between healthy controls and subjects with some neuropathology, such an approach could also be employed in the context of task-based fMRI as well as to study changes in functional connectivity induced by various distinct tasks [54] or neuropathologies [55, 56]. In particular, task-based fMRI has been widely reported as displaying non-stationary functional connectivity structure [57, 58, 59, 60]. As such, seeking to discriminate between various cognitive tasks, for example as considered by [61], [62], [63, 64], could be an exciting future application. Moreover, while in this work we have considered linear latent variable models such as PCA, future work could consider alternative latent variable modes such as latent position graphs [65] and causal models [66, 67, 68].

## Supporting information

**S1 Appendix. Technical details of the MHA algorithm.**
(PDF)

**S1 Code. Python and R implementations of the MHA algorithm.**
(PDF)

**S1 Table. Table detailing number of subjects studied in each of the three datasets considered.** In the case of the HCP datasets, 80 subjects were randomly selected out of all possible

subjects.
(PDF)

**S1 Fig. Age distributions of subjects across repositories.**
(PNG)

**S2 Fig. Functional connectivity networks inferred by PCA and alternative models.**
(PARTIAL)

**S3 Fig. Generalization performance of brain age prediction on HCP and ATR Wide-Age-Range datasets.**
(PNG)

## Acknowledgments

The authors with to thank Steve Smith for valuable feedback and discussions.

## Author Contributions

**Conceptualization:** Ricardo Pio Monti, Alex Gibberd, Sandipan Roy, Matthew Nunes, Romy Lorenz, Robert Leech, Motoaki Kawanabe, Aapo Hyvärinen.

**Data curation:** Ricardo Pio Monti, Romy Lorenz, Robert Leech, Takeshi Ogawa, Motoaki Kawanabe, Aapo Hyvärinen.

**Formal analysis:** Ricardo Pio Monti, Sandipan Roy, Matthew Nunes, Romy Lorenz, Motoaki Kawanabe, Aapo Hyvärinen.

**Funding acquisition:** Robert Leech, Motoaki Kawanabe, Aapo Hyvärinen.

**Investigation:** Ricardo Pio Monti, Alex Gibberd, Sandipan Roy, Matthew Nunes, Romy Lorenz, Robert Leech, Motoaki Kawanabe, Aapo Hyvärinen.

**Methodology:** Ricardo Pio Monti, Alex Gibberd, Sandipan Roy, Matthew Nunes, Romy Lorenz, Robert Leech, Motoaki Kawanabe, Aapo Hyvärinen.

**Project administration:** Ricardo Pio Monti, Romy Lorenz, Motoaki Kawanabe, Aapo Hyvärinen.

**Resources:** Ricardo Pio Monti, Motoaki Kawanabe, Aapo Hyvärinen.

**Software:** Ricardo Pio Monti, Motoaki Kawanabe, Aapo Hyvärinen.

**Supervision:** Motoaki Kawanabe, Aapo Hyvärinen.

**Validation:** Ricardo Pio Monti, Alex Gibberd, Sandipan Roy, Matthew Nunes, Romy Lorenz, Robert Leech, Takeshi Ogawa, Motoaki Kawanabe, Aapo Hyvärinen.

**Visualization:** Ricardo Pio Monti, Alex Gibberd, Sandipan Roy, Matthew Nunes, Romy Lorenz, Robert Leech, Takeshi Ogawa, Motoaki Kawanabe, Aapo Hyvärinen.

**Writing – original draft:** Ricardo Pio Monti, Alex Gibberd, Sandipan Roy, Matthew Nunes, Romy Lorenz, Robert Leech, Takeshi Ogawa, Motoaki Kawanabe, Aapo Hyvärinen.

**Writing – review & editing:** Ricardo Pio Monti, Alex Gibberd, Sandipan Roy, Matthew Nunes, Romy Lorenz, Robert Leech, Takeshi Ogawa, Motoaki Kawanabe, Aapo Hyvärinen.

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
