## [Decision Letter · Decision Letter 0]

11 Feb 2020

PONE-D-19-33576

Interpretable  brain age prediction using linear latent variable models of functional connectivity

PLOS ONE

Dear Dr Monti,

Thank you for submitting your manuscript to PLOS ONE. After careful consideration, we feel that it has merit but does not fully meet PLOS ONE’s publication criteria as it currently stands. Therefore, we invite you to submit a revised version of the manuscript that addresses the points raised during the review process.

We would appreciate receiving your revised manuscript by Mar 27 2020 11:59PM. To enhance the reproducibility of your results, we recommend that if applicable you deposit your laboratory protocols in protocols.io, where a protocol can be assigned its own identifier (DOI) such that it can be cited independently in the future. For instructions see: http://journals.plos.org/plosone/s/submission-guidelines#loc-laboratory-protocols

We look forward to receiving your revised manuscript.

Kind regards,

Carlo Vittorio Cannistraci

Academic Editor

PLOS ONE

Additional Editor Comments (if provided):

Dear Authors

please address carefully all the comments advanced by the Reviewer

thanks

Carlo Vittorio Cannistraci

Journal Requirements:

2. Please ensure that you refer to Figure 12, 13, 14 and 15 in your text as, if accepted, production will need this reference to link the reader to the figure.

3. We note that Figures 1, 2, 5 and 13 in your submission contain copyrighted images. All PLOS content is published under the Creative Commons Attribution License (CC BY 4.0), which means that the manuscript, images, and Supporting Information files will be freely available online, and any third party is permitted to access, download, copy, distribute, and use these materials in any way, even commercially, with proper attribution. For more information, see our copyright guidelines: http://journals.plos.org/plosone/s/licenses-and-copyright.

1.    You may seek permission from the original copyright holder of Figures 1, 2, 5 and 13 to publish the content specifically under the CC BY 4.0 license.

Reviewers' comments:

Reviewer's Responses to Questions

**Comments to the Author**

1. Is the manuscript technically sound, and do the data support the conclusions?

Reviewer #1: Yes

2. Has the statistical analysis been performed appropriately and rigorously? 

Reviewer #1: Yes

3. Have the authors made all data underlying the findings in their manuscript fully available?

Reviewer #1: Yes

4. Is the manuscript presented in an intelligible fashion and written in standard English?

Reviewer #1: Yes

5. Review Comments to the Author

Reviewer #1: The work proposed a very interesting a powerful analytical framework to study the aging dynamics of human functional connectivity. The mathematical presentation is flawless and clearly presented. Interestingly, the method appeared extendable to different contexts studying the dynamical aspects of functional connectivity, a relevant topic to date. First of all, authors should consider this aspect in the discussion/conclusion section. Moreover, Interested readers will find all details in order to reproduce results. However, I have some complains that authors should accomplish upon acceptance of the work:

1. Although authors use HCP Young Adult dataset just for test, they should declare the number of subject used.

2. Most importantly, within the human connectome project, there exists a similar collection called "HCP Aging"

chracterized by 1200 Subjects in the age range of 36-100+ years old. That's the dataset they should test.

3. If, the python "plot_glass_brain" function has been used to plot figures 5 and S4 (as I assumed), they should state it because otherwise it is necessary to specify the x-y-z coordinates. That function put in foreground every network

elements (nodes/edges) and the brain in background and it is particularly useful in displaying brain network.

4. However,

authors stated (in captions and text) those plots as "networks" but just nodes (ROI centroids?) are presented. This discrepancy should be fixed.

6. PLOS authors have the option to publish the peer review history of their article (what does this mean?). If published, this will include your full peer review and any attached files.

Reviewer #1: Yes: Antonio Giuliano Zippo

---

## [Author Response · Author response to Decision Letter 0]

12 Mar 2020

We attach a detailed response to reviewers.

---

## [Decision Letter · Decision Letter 1]

13 Apr 2020

Interpretable  brain age prediction using linear latent variable models of functional connectivity

PONE-D-19-33576R1

Dear Dr. Monti,

We are pleased to inform you that your manuscript has been judged scientifically suitable for publication and will be formally accepted for publication once it complies with all outstanding technical requirements.

With kind regards,

Carlo Vittorio Cannistraci

Academic Editor

PLOS ONE

Additional Editor Comments (optional):

Reviewers' comments:

Reviewer's Responses to Questions

**Comments to the Author**

1. If the authors have adequately addressed your comments raised in a previous round of review and you feel that this manuscript is now acceptable for publication, you may indicate that here to bypass the “Comments to the Author” section, enter your conflict of interest statement in the “Confidential to Editor” section, and submit your "Accept" recommendation.

Reviewer #1: All comments have been addressed

2. Is the manuscript technically sound, and do the data support the conclusions?

Reviewer #1: Yes

3. Has the statistical analysis been performed appropriately and rigorously? 

Reviewer #1: Yes

4. Have the authors made all data underlying the findings in their manuscript fully available?

Reviewer #1: Yes

5. Is the manuscript presented in an intelligible fashion and written in standard English?

Reviewer #1: Yes

6. Review Comments to the Author

Reviewer #1: (No Response)

7. PLOS authors have the option to publish the peer review history of their article (what does this mean?). If published, this will include your full peer review and any attached files.

Reviewer #1: Yes: Antonio Giuliano Zippo

---

## [Editor Report · Acceptance letter]

13 May 2020

PONE-D-19-33576R1 

Interpretable  brain age prediction using linear latent variable models of functional connectivity 

Dear Dr. Monti:

I am pleased to inform you that your manuscript has been deemed suitable for publication in PLOS ONE. Congratulations! Your manuscript is now with our production department. 

With kind regards,

on behalf of

Dr. Carlo Vittorio Cannistraci 

Academic Editor

PLOS ONE